

# Seasonal dynamics in leaf litter decomposing microbial communities in temperate forests: a whole-genome-sequencing-based study

Nataliia Khomutovska[1,2], Iwona Jasser[2], Polina Sarapultseva[3], Viktoria Spirina[3], Andrei Zaitsev[4], Jolanta Masłowiecka[5] and Valery A. Isidorov[5]

[1] Department of Plant Protection Biology, Swedish University of Agricultural Sciences, Lomma, Skane, Sweden, Lomma, Sweden
[2] Department of Ecology and Environmental Conservation, Faculty of Biology, Biological and Chemical Research Centre, University of Warsaw, Warsaw, Poland
[3] Chemical Department of Perm, State University, Perm, Russia
[4] Faculty of Geography of Perm, State University, Perm, Russia
[5] Institute of Forest Sciences, Białystok University of Technology, Białystok, Poland

Corresponding author
Nataliia Khomutovska,
n.khomutovska@uw.edu.pl

## ABSTRACT

Leaf litter decomposition, a crucial component of the global carbon cycle, relies on the pivotal role played by microorganisms. However, despite their ecological importance, leaf-litter-decomposing microorganism taxonomic and functional diversity needs additional study. This study explores the taxonomic composition, dynamics, and functional role of microbial communities that decompose leaf litter of forest-forming tree species in two ecologically unique regions of Europe. Twenty-nine microbial metagenomes isolated from the leaf litter of eight forest-forming species of woody plants were investigated by Illumina technology using read- and assembly-based approaches of sequences analysis. The taxonomic structure of the microbial community varies depending on the stage of litter decomposition; however, the community's core is formed by *Pseudomonas*, *Sphingomonas*, *Stenotrophomonas*, and *Pedobacter* genera of Bacteria and by *Aureobasidium*, *Penicillium*, *Venturia* genera of Fungi. A comparative analysis of the taxonomic structure and composition of the microbial communities revealed that in both regions, seasonal changes in structure take place; however, there is no clear pattern in its dynamics. Functional gene analysis of MAGs revealed numerous metabolic profiles associated with leaf litter degradation. This highlights the diverse metabolic capabilities of microbial communities and their implications for ecosystem processes, including the production of volatile organic compounds (VOCs) during organic matter decomposition. This study provides important advances in understanding of ecosystem processes and the carbon cycle, underscoring the need to unravel the intricacies of microbial communities within these contexts.

## INTRODUCTION

Leaf litter is a transition zone between soil and atmosphere. Litter decomposition is one of the main elements of the global carbon and nutrient cycle (*Krishna & Mohan, 2017*; *Fontaine et al., 2007*). Microorganisms, fungi and bacteria play leading roles in this process (*Djukic et al., 2018*; *Trowbridge, Stoy & Phillips, 2020*), during which they produce metabolites, including volatile organic compounds (VOCs), which can pass into the gas phase (*Trowbridge, Stoy & Phillips, 2020*; *Bonanomi et al., 2017*; *Leff & Fierer, 2008*; *Svendsen et al., 2018*). Biogenic emission of VOCs is particularly important due to their essential roles in many global processes, including the oxidative potential of the atmosphere, production of secondary organic aerosols, the radiation regime of the troposphere and, consequently, global climate (*Isidorov, 1990*; *Koppmann, 2008*). The role of individual VOCs in these processes is determined by their chemical structure and the scale of their emission into the atmosphere. The VOC emission from decayed leaf litter is a blend of plant tissue-derived volatiles and compounds newly enzymatically synthesized by microbial communities. The chemical composition of the latter is species-specific (*Isidorov, Tyszkiewicz & Pirożnikow, 2016*; *Isidorov & Zaitsev, 2022*); however, the role of specific taxa of microorganisms in the decomposition of litter and the contribution of bacterial and fungal species to the emission of VOCs needs more study (*Isidorov & Zaitsev, 2022*; *Cox, Wilkinson & Anderson, 2001*; *Fioretto et al., 2007*; *Hobbie et al., 2012*; *Kirker et al., 2020*; *Frankland, 1998*).

Fungi are believed to play a major role in the degradation of litter: they can assimilate readily available compounds and degrade chemically stable biopolymers (cellulose, lignocellulose and lignin) through the production of appropriate enzymes (*Svendsen et al., 2018*; *Koppmann, 2008*; *Isidorov, Tyszkiewicz & Pirożnikow, 2016*; *Isidorov & Zaitsev, 2022*; *Cox, Wilkinson & Anderson, 2001*). Therefore, much attention has been paid to the species composition of fungi, their changes at different stages of litter decomposition and the composition of their volatile metabolites (*Cox, Wilkinson & Anderson, 2001*; *Fioretto et al., 2007*; *Hobbie et al., 2012*; *Kirker et al., 2020*). Litter-degrading bacteria and their associated VOC production have received less attention, although their involvement in these processes is well documented (*Djukic et al., 2018*; *Frankland, 1998*).

Several studies have shown significant differences in the dynamics of microbial communities at different stages of litter decomposition (*Silva-Sanchéz, Soares & Rousk, 2019*; *Soares, Kritzberg & Rousk, 2017*; *Schroeter et al., 2022*). However, until now, most work on microbial leaf-litter decomposing communities has used amplicon-sequencing methods. Ergosterol examination (*Soares, Kritzberg & Rousk, 2017*) or phylogenetic marker gene-based analysis (such as 16S ribosomal RNA [rRNA], 18S rRNA and the internal transcribed spacer (ITS)) are important if the main purpose is to identify microorganisms (*Schroeter et al., 2022*). However, analysis of these amplicons does not allow functional exploration of the microbial community (*Schroeter et al., 2022*).

A better understanding of the functioning of the leaf litter decomposing ecosystem and the metabolic potential of its communities in temperate and boreal zones could be provided by whole-metagenome-based examination of microorganisms.

Previous research on leaf litter microbial communities primarily focused on cultured microorganisms or short amplicon sequences (*Purahong et al., 2016*). These studies highlight microbial community specialization based on the type of leaf litter substrate (*Schroeter et al., 2022*; *Purahong et al., 2016*; *Veen et al., 2021*). There can be differences in the bacterial communities, even between those that decompose the leaf litter of two birch species (*Betula pendula* and *Betula pubescens*), as shown by the composition of synthesized secondary metabolites (*Isidorov et al., 2014*; *Isidorov et al., 2021*). Such differences may influence the composition of microbial communities involved in litter decomposition. Leaf litter of other forest-forming deciduous trees, including aspen (*Populus tremula*) and hornbeam (*Carpinus betulus*), can create more specific microenvironmental conditions due to the chemicals produced during decomposition, which could be the cause of the development of unique microbiomes. In contrast to taxonomic diversity, extensively studied in environments similar to those investigated within this work, the functional potential of microbial communities (assessed through functional gene analysis) remains relatively unexplored, particularly using metagenomic tools. The limited research on functional diversity includes examination of the role of microorganisms in nitrogen (N) cycling, utilizing various assimilatory and dissimilatory pathways. The genetic potential of N cycle processes in plant litter is a blueprint for understanding ecosystem nutrient dynamics. Previous studies have shown significant differences in the frequencies of sequences associated with eight N cycling pathways among microbial taxa, indicating varying levels of specialization and generalization within bacterial orders. Furthermore, temporal variability and environmental stressors such as drought influence microbial N cycling gene abundance and composition, highlighting the dynamic response of microbial communities to changing environmental conditions (*Nelson, Martiny & Martiny, 2015*). Similarly, investigations into microbial communities in Californian grass and shrub ecosystems under long-term drought reveal distinct taxonomic and functional profiles. Metatranscriptomic and metabolomic analyses elucidate physiological responses, such as producing compatible solutes and extracellular polymeric substances, indicating microbial adaptation strategies to drought stress. These findings underscore the importance of understanding the trade-offs between microbial stress tolerance, resource acquisition, and growth traits in shaping decomposition processes under changing environmental conditions (*Malik et al., 2020*). Furthermore, network analysis of metagenomic data from mangrove litter decomposition reveals robust and stable microbial communities resilient to environmental pressures and substrate variability. This study sheds light on the complex interactions between microbial populations, litter chemical composition, and environmental parameters, providing insights into ecosystem functioning and stability (*Taketani et al., 2010*). Even using an amplicon-based approach and chemical analysis, we may overlook the presence of rare genes and information concerning the presence of taxa that could be important for the biogeochemical process. Thus, this work focused on whole-metagenome analysis to comprehensively investigate the genetic potential of litter-decomposing microbial communities.

Recent studies (*Silva-Sanchéz, Soares & Rousk, 2019*; *Soares, Kritzberg & Rousk, 2017*; *Schroeter et al., 2022*) have highlighted significant differences in microbial community

dynamics at different stages of litter decomposition, underscoring the need for a more comprehensive understanding of these processes from an ecological perspective. Seasonal variations in environmental factors such as temperature, precipitation, and nutrient availability can profoundly affect microbial communities, potentially overshadowing the influence of specific litter types (*Schroeter et al., 2022*).

While litter composition provides essential carbon sources and substrates for microbial metabolism, seasonal fluctuations in environmental conditions, such as temperature and moisture levels, can significantly modulate microbial activity and community structure (*Schroeter et al., 2022*). Therefore, our hypotheses are framed within litter characteristics and seasonal dynamics, acknowledging the complex interplay between these factors in shaping microbial community dynamics during litter decomposition. Informed by our review of existing literature and preliminary investigations, our study contributes valuable insights into the understudied aspects of microbial ecology in leaf litter decomposition ecosystems. By integrating taxonomic and functional analyses, we aim to advance our understanding of the complex interactions between microbial communities and their environment, with implications for ecosystem functioning and resilience in the face of environmental change (*Koivusaari et al., 2019*; *Hartikainen et al., 2009*; *Yakimovich et al., 2018*; *Soares & Rousk, 2019*; *Wahdan et al., 2021*; *Hu et al., 2022*; *Lu et al., 2022*).

Our first hypothesis proposes that seasonal changes in both micro- and macro-environments shape microbial community composition more than the specific type of leaf litter. If proven, this would suggest a universal presence of certain core microbial taxa in leaf litter environments, regardless of geographic location. Our second hypothesis proposes that the metabolic potential of microbial communities is influenced by both the type of leaf litter and its stage of decomposition. This would indicate that leaf litter characteristics and decomposition stages can affect the functional profiles of microbial communities, even across geographically distant regions.

Our study aims to: (i) Utilize whole-genome sequencing to examine the composition and structure of bacterial and fungal communities. (ii) Investigate how microbial community structure changes over time across different seasons and stages of decomposition in temperate and boreal forests spanning two geographically distant regions. (iii) Characterize the ecophysiological capabilities of litter-decomposing bacteria and fungi, focusing on gene clusters associated with volatile secondary metabolites. Due to the limited data on leaf litter decomposing microorganisms from the studied regions, our research mainly explores bacterial and fungal community composition, structure and functional diversity, providing valuable insights into these understudied aspects of microbial ecology.

## MATERIALS & METHODS

### Study area, leaf litter incubation and sampling

For our analysis, we purposefully selected two regions characterized by distinct and contrasting environmental conditions, providing an ideal setting to investigate the intricate processes of leaf litter decomposition. The investigated areas are situated in northern Poland (Kopna Góra arboretum) and Russia ("Osinskaya dacha" forest, Perm region). The

Kopna Góra arboretum belongs to a large complex of the Puszcza Knyszynska Forest and is located 28 km to the north-east of Białystok (53°14′N and 23°29′E). The Kopna Góra arboretum is on a flat area of deep glaciofluvial sand sediments, 135 m above sea level. The mean annual precipitation is 650 mm (1993–2022), and mean annual temperature is 7.1 °C. The duration of the growing season is about 200 days. The tree layer is comprised of 50–60-year-old *Pinus sylvestris* and *Picea abies*. The ground vegetation, completely covering the ground, is composed mainly of the mosses *Pleurozium schreberi* (Brud.) Mitt. and *Dicranum polysetum* Sv., together with lingonberry, *Vaccinium vitis-idaea* L. In the Perm region of Russia, material was collected within the enormous forest massif of pine and mixed forests of the "Osinskaya dacha" forest. The territory is located within the subzone of coniferous-deciduous forests on the eastern edge of the Russian Plain (geographical coordinates 57°28′N and 55°16′E). The phytocoenosis is represented by pine forests formed on dunes. *Pleurozium schreberi* and species of the genus *Dicranum* are widespread in the mossy-lichen tier. Blueberry (*Vaccinium myrtíllus*) and lingonberry (*Vaccinium vítis-idaéa*) are widely represented in the herbaceous-shrub layer.

The experiment utilized incubation plots composed of terylene mesh with a mesh size of 1.5 mm, forming the bottom of litter bags (200 × 200 × 20 mm). Samples were collected from the litter bags from both regions mainly in autumn, *i.e.,* October–November, and late spring, *i.e.,* May. At the beginning stage of the experiment, fresh fallen leaves of trees from litter of two species of birch (*Betula pendula* and *B. pubescens*), aspen (*P. tremula*), hornbeam (*C. betulus*), English oak (*Quercus robur*), littleleaf linden (*Tilia cordata*), Norway maple (*Acer platanoides)*, and black alder (*Alnus glutinosa*) were utilized as the initial material. These leaves gradually decomposed over subsequent stages of decomposition facilitated by microorganisms (Fig. S1). In this study, materials were incubated under natural environmental conditions from the beginning of the experiment until reaching an incubation period of 18 months. The leaf materials underwent a gradual decomposition process facilitated by a consortium of microorganisms, influenced by seasonal fluctuations in temperature, humidity, and the natural microbial activity present in the outdoor environment. Additional sampling was done in February only from the study area in Poland. The experiment was described previously (*Isidorov, Maslowiecka & Sarapultseva, 2024*). For metagenomics analysis, leaf litter was collected on sterile Petri dishes that were delivered to the lab and dried for DNA extraction and further analysis.

## Sample preparation, DNA extraction and sequencing

Before DNA extraction, litter was homogenized using sterile hand homogenizers with liquid nitrogen. For DNA extraction, commercially developed kits such as the E.Z.N.A.® Soil DNA Kit (Omega Bio-tek, Norcross, GA, USA) and the Soil DNA Purification Kit (GeneMATRIX, EURx Ltd., Gdańsk, Poland) were used. The procedure was optimized as in our previous work (*Khomutovska et al., 2020*; *Khomutovska, Delos Ríos & Jasser, 2021*). For each sample, DNA extraction was performed 5–10 times depending on the amount and quality of the DNA obtained. Leaf-litter samples collected at the first phase of decomposition were most problematic due to the higher content of host DNA. Library preparation and whole-metagenome sequencing was performed commercially by Eurofins Genomics AT

GmbH using the Illumina platform (NovaSeq 6000 S4 PE150 XP sequence mode, targeting 20M reads per sample). Illumina raw reads were submitted under BioProject ID: 1001592.

## Bioinformatics and statistical analysis

The bioinformatic analysis was conducted using the Funk server of the Biological and Chemical Centre of the University of Warsaw. The quality control of the paired-end reads was performed using the FASTQC tool (*Andrews, 2010*). The Cutadapt tool was used to trim the raw read (*Martin, 2011*). This study employed two strategies: read-based analyses using DIAMOND+MEGAN v.6. (*Buchfink, Xie & Huson, 2015*; *Huson et al., 2007*) and an assembly approach using MEGAHIT (*Li et al., 2015*; *Li et al., 2016*). The plant sequences were discarded, and the dataset was normalized for further analysis. The reference databases used were the National Center for Biotechnology Information (NCBI) (*Altschul et al., 1997*) and the Genome Taxonomy Database (GTDB) (*Parks et al., 2021*). Functional gene annotation of assembles and MAGs (metagenome-assembled genomes) was conducted utilizing distilled and refined annotation of metabolism (DRAM) software (*Shaffer et al., 2020*), facilitating the characterization of metabolic pathways within the microbial communities under investigation. Binning of the metagenomic data was executed employing MaxBin2 (*Wu, Simmons & Singer, 2016*), followed by quality control assessment utilizing QUAST (*Gurevich et al., 2013*) and optimization procedures using dRep (*Olm et al., 2017*), ensuring the robustness and accuracy of the obtained MAGs. The MAGs were mapped on the phylogenetic tree using KBase (*Chivian et al., 2023*). The phylogenetic analysis of obtained MAGs was conducted by aligning the MAG sequences with a curated set of thoroughly characterized genomes from the reference database. After filtering out plant DNA, we obtained sufficient data for the characterization of microbial communities in our samples, as supported by previous studies (*Pérez-Cobas, Gomez-Valero & Buchrieser, 2020*).

Furthermore, statistical analyses, including diversity calculation and visualization, were carried out using the R project (*R Core Team, 2018*) framework, with specific packages such as heatmap (*Kolde, 2019*) and vegan (*Oksanen et al., 2013*; *Anderson & Walsh, 2013*).

## RESULTS

### Taxonomic composition of litter-decomposing microbial communities

The present whole-metagenome-based study describes the taxonomic composition of microbial communities decomposing leaf litter of two species of birch (*B. pendula* and *B. pubescens*), aspen (*P. tremula*), hornbeam (*C. betulus*), English oak (*Quercus robur*), littleleaf linden (*Tilia cordata*), Norway maple (*Acer platanoides*) and black alder (*Alnus glutinosa*) (Fig. 1). The heatmaps present normalized reads per sample (Figs. 2 and 3).

According to the NCBI database, 2,820,491 assigned sequences passed quality filters (Fig. n2, Table S1). Read-based analysis of the taxonomic profile based on the NCBI database revealed the dominance of bacteria in most samples. Eukaryota (Fungi) prevailed in only eight samples: AP12_R (Acer 12 months of decomposition), PT12_R (aspen 12 months of exposure), TP12_R, QR0_P, PT0_P, CB12_P, CB0_P and BP18_P (Fig. 2). Trace amounts of viral and archaeal sequences were present in the samples. They varied from 27

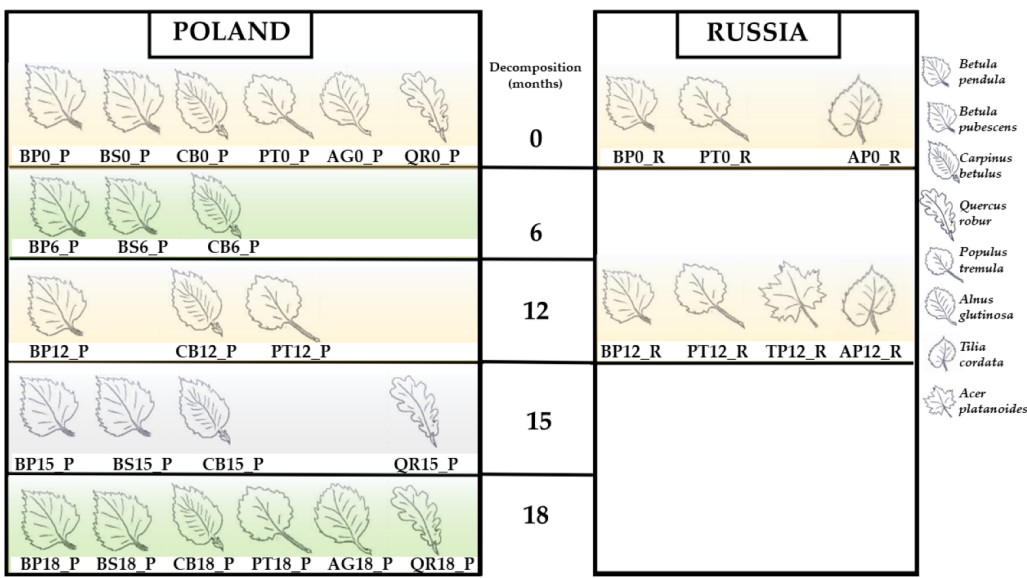

**Figure 1** **Studied samples of leaf litter.** *Betula pendula*—BP; *Betula pubescens*—BS; *Carpinus betulus*—CB; *Populus tremula*—PT; *Alnus glutinosa*—AG; *Quercus robur*—QR; *Tilia cordata*—TP; *Acer platanoides*.

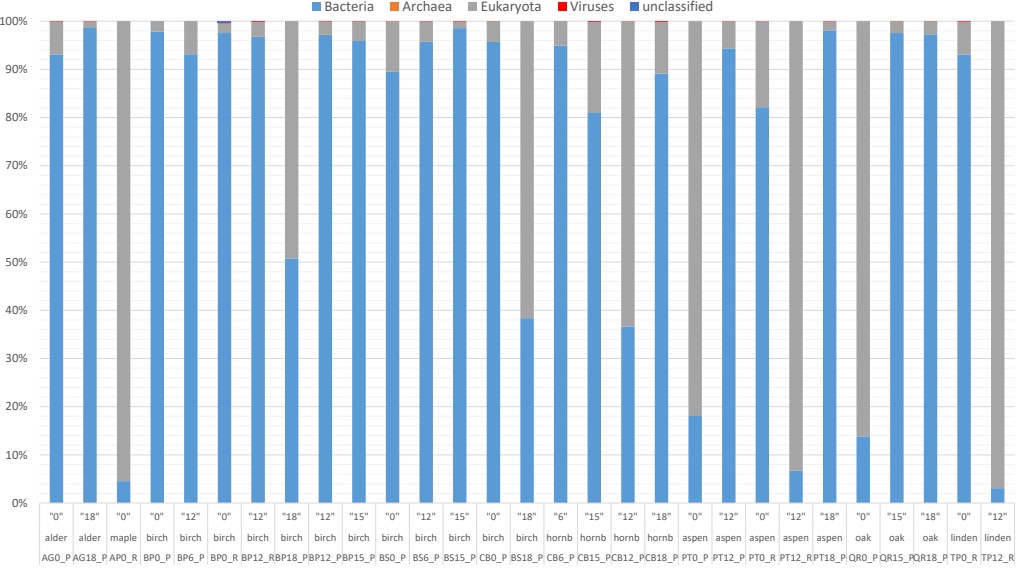

**Figure 2** **Taxonomic composition based on the National Center for Biotechnology Information (NCBI) taxonomy for the entire community.** Read-based metagenomic analysis of the composition of bacterial communities.

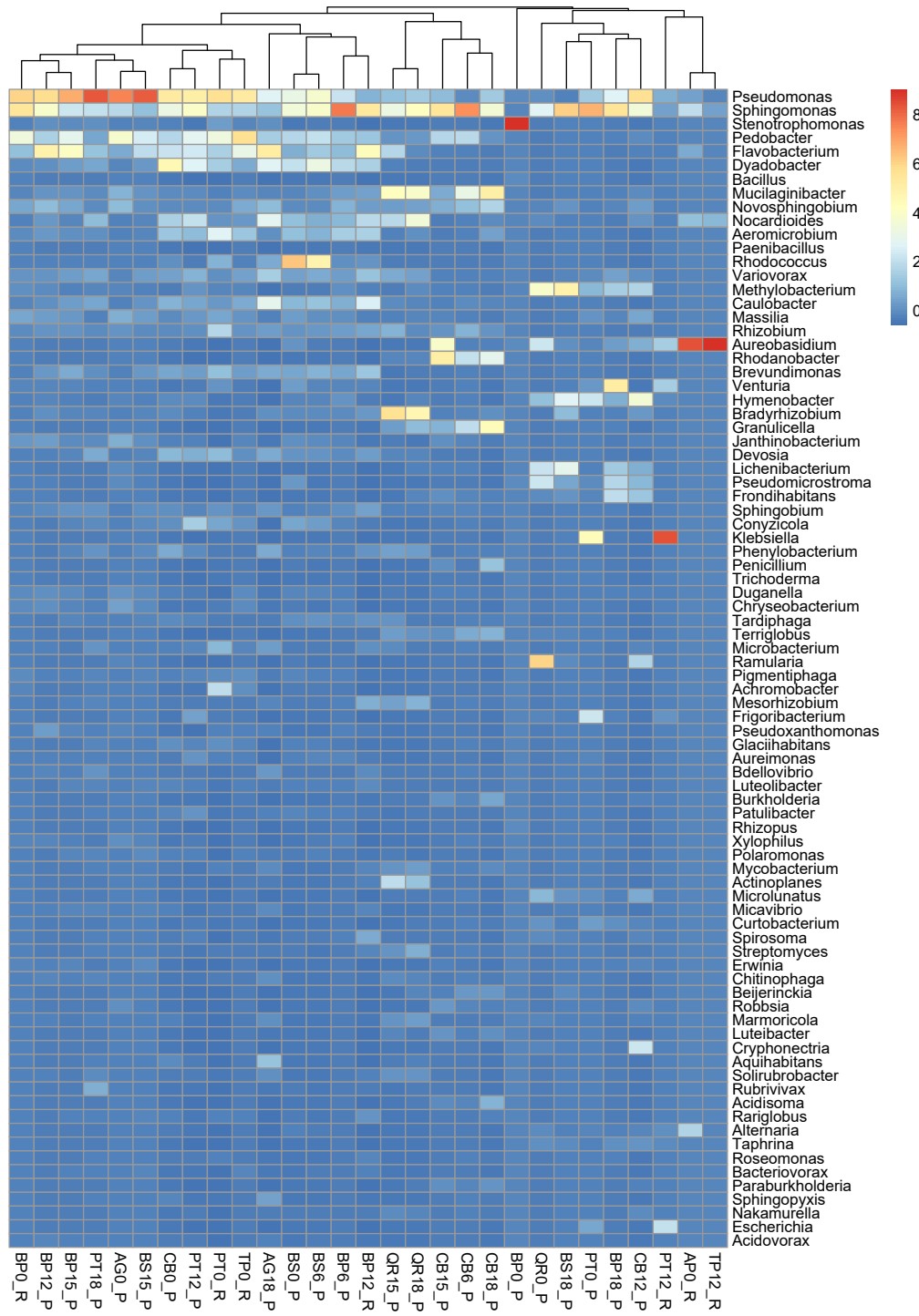

**Figure 3  Taxonomic composition and structure of microbial communities at the genus level.** Read-based analysis. Scaled dataset.

(BP6_P) to 271 (PB0_R) and from 16 (PT18_P) to 51 (CB15_P), respectively, for viruses and archaea (Fig. 2). Viral and archaeal sequences were not detected in the metagenomes of silver birch (BP0_P), oak (QR0_P), aspen (PT0_P) and hornbeam (CB0_P) from Poland and maple (AP12_R4, PT12_R, TP12_R). According to the NCBI database, the highest number of unclassified sequences was observed in the birch sample (BS15_P, 16 reads).

According to NCBI classification, the most abundant group of microorganisms was Pseudomonadota, whose number of sequences varied from 892 (TP12_R) to 75,877 (BP0_P). Generally, the percentage of Pseudomonadota was high in the late stage of decomposition (15–18 months). The second most abundant group was Bacteroidota, whose percentage fluctuated from < 1% (15 reads, BP0_P) to ∼30% (24,037 reads, TP0_R). The third most abundant group of microorganisms were fungi, represented mainly by Ascomycota, varying from a few per cent (597 reads, AG18_P) to > 50% (45,062 reads, CB12_P).

The percentage of Actinobacteriota was higher at the earlier stage of leaf litter decomposition in most of the studied metagenomes. The number of sequences varied from < 0.01% (65, BP0_P) to > 25% (23,096, PT12_P).

Viruses were not abundant in the metagenomes. They were represented mainly by Uroviricota, with 22–242 sequences (Fig. 2).

Burkholderiales dominates in several samples, it has the highest number of reads in sample PT0_R. Sphingomonadales was prominent in samples most of the samples including those from Poland and Russia collected at the begining of decomposition experiment ass and latest stages (15 and 18 months). Among these, it has the highest number of reads in sample of birch from Russia (BP12_R). Sphingobacteriales dominates in samples most of studied samples, it has the highest number of reads in sample AG0_P. Flavobacteriales, Cytophagales, and Caulobacterales show varying dominance across different samples, but generally have fewer reads than the above orders. Other orders, such as Acidobacteriales, Enterobacterales, and Rhodospirillales, also appear, but with lower read counts across the samples (Fig. S2).

*Pseudomonas* exhibits high abundance across multiple samples, particularly prevalent in alder and downy birch litter (0 moths of decomposition). *Sphingomonas* were abundant in samples like alder (AG0_P), maple from Russia (AP16_R), silver birch from Poland (BP0_P, and BP15_P), with relatively lower presence in other samples. *Stenotrophomonas* were predominant in samples of silver birch (BP6_P) from Poland and aspen from Russia (PT0_R), with minimal presence in other samples. *Pedobacter* is commonly found in samples of alder (AG0_P), maple (AP16_R), and downy birch (BS0_P, BS15_P). *Flavobacterium* was well-represented in alder (AG0_P, AG18_P), and silver birch (BP0_P) samples, with lower abundance in others. *Rhodococcus* is abundant in samples of birch and hornbeam from Poland (BS0_P, BS15_P, and CB0_P). *Variovorax* was found in various samples but notably abundant in in samples of birch and hornbeam. *Rhizobium, Methylobacterium, Brevundimonas Massilia* and *Caulobacter* are is commonly found in samples of alder and maple (AG0_P, AG18_P, AP16_R), and downy birch (BS0_P) samples. *Aureobasidium* is particularly abundant in alder and downy birch samples. *Rhodanobacter* is mostly found in aspen samples collected from Russia and Poland (Fig. 3).

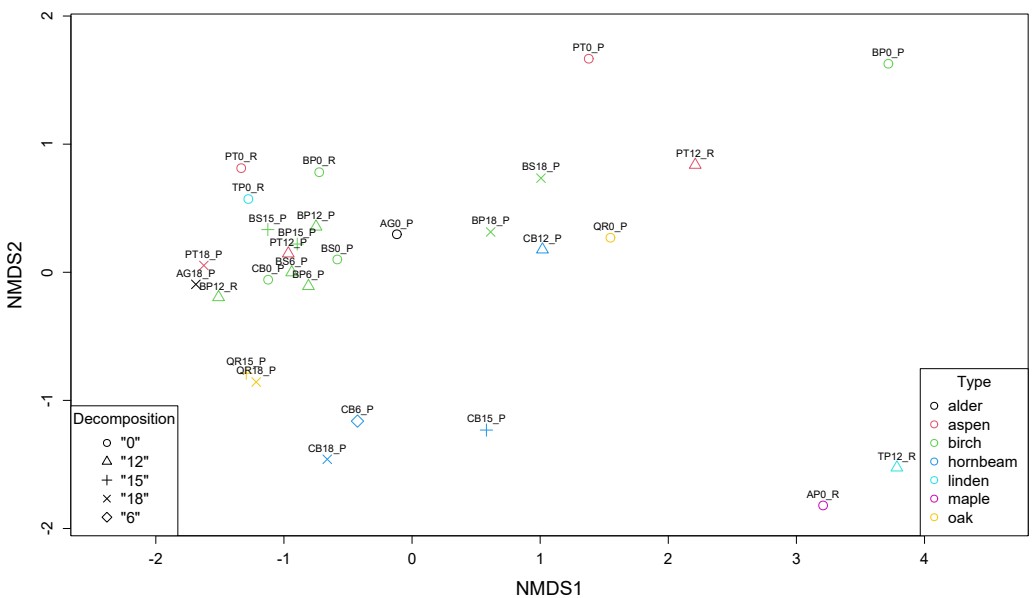

**Figure 4  Comparison of the structure of leaf litter decomposing microbial communities.**

Alpha diversity assessed for the entire microbial community revealed that the highest values of the Shannon diversity index were for samples from Poland, including birch in the later stages of decomposition silver (BP12_P–6.46,), and downy birch (BS6_P–6.5, BS15_P–6.6), black alder (AG18_P–6.6) and aspen (PT12_P–6.5) (Table S2).

Non-metric multidimensional scaling (NMDS) revealed the presence of three main clusters of metagenomes. The first cluster of microbial communities includes metagenomes isolated from birch (BP0_P, BP18_P, BS18_P), hornbeam (CB18_P, CB15_P, CB12_P), and aspen (PT0_P, PT12_R), while the second and the largest group gathers most of the birch samples accompanied by two communities of aspen (PT12_P and PT0_P) and one of alder (AG0_P). There are two other more variable clusters (Fig. 4).

Our analysis corresponds with the Adonis (PERMANOVA) test outcomes, indicating findings regarding leaf litter composition and microbial community structure. The significant impact of the type of leaf litter variable on the distance matrix, indicated by a low $p$-value (0.006), underscores the importance of leaf litter composition in shaping microbial community structure during decomposition phases. This finding suggests that variations in the type of leaf litter significantly influence microbial diversity and composition. On the other hand, the decomposition phase as a variable representing the decomposition phase did not exhibit a statistically significant effect on the distance matrix, with a $p$-value of 0.566 exceeding the significance level of 0.05. While decomposition stages may influence microbial activity and function, their impact on taxonomic structure is less pronounced in our study.

Comparative analysis of microbial abundance in leaf litter revealed distinct patterns across various tree species. *Pseudomonas*, *Sphingomonas*, and *Stenotrophomonas* were prevalent in specific litter types, while *Pedobacter*, *Flavobacterium*, *Rhodococcus*, *Variovorax*,

and others showed varying levels of abundance across different litter samples. Birch samples from Poland exhibited the highest Shannon diversity index values, indicating greater microbial diversity.

Furthermore, the NMDS identified three main clusters of metagenomes, suggesting distinct microbial community structures associated with different leaf litter compositions. The Adonis (PERMANOVA) test supported these findings, indicating an association between leaf litter composition and microbial community structure during decomposition phases. This emphasizes the influence of leaf litter type on microbial diversity and suggests that microbial abundance varies in response to leaf litter composition, shaping the overall microbial community structure during decomposition.

### Functional genes analysis of leaf-litter decomposing communities

Metabolic pathway analysis reveals the following samples with diverse representations, showcasing the metabolic versatility within the microbial communities studied: PT0_R shows the potential for a wide range of metabolic activities or functional capacity within the microbial community (reductive citrate cycle, glycolysis, Entner-Doudoroff pathway). PT18_P shows a significant presence of metabolic pathways akin to PT0_R and BP18_P; this sample implies a diverse range of metabolic activities within the microbial community (glycolysis, reductive pentose phosphate cycle, methanogenesis). PT12_P, with slightly fewer pathways represented, still has notable diversity, suggesting active metabolic processes and functional versatility within the microbial population (glycolysis, TCA cycle, and reductive pentose phosphate cycle). The PERMANOVA analysis reveals significant contributions from both types of leaf litter and decomposition factors to the variation in functional gene diversity. The leaf litter type explains a substantial portion of the variation ($R^2 = 0.87007$) with a near-significant F-statistic of 3.8872 ($p = 0.077$). Similarly, the decomposition phase also contributes significantly ($R^2 = 0.05924$) with an F-statistic of 5.0288 ($p = 0.081$). The residual variation, unexplained by the model, remains relatively low ($R^2 = 0.07068$) (Figs. 5, S2).

We conducted functional gene analysis for optimized with dRep MAGs. We carefully evaluated the quality of metagenome-assembled genomes (MAGs) by checking parameters like average nucleotide identity (ANI). Only MAGs meeting quality standards were included in our analysis, ensuring our results were based on reliable genomic data. Our results revealed distinct metabolic profiles associated with leaf litter degradation in specific microbial MAGs. For instance, the acetyl-CoA pathway was fully present in MAGs like 002.BP12_R, 002.BS6_P, 003.BP12_R and CB15_P, while others showed no indication of it.

Identified MAGs belong to *Brevundimonas* (BS15_P), *Rhodococcus* (BP6_P), *Sphingomonas* (BP6_P), *Telluria* (BS0_P, BS6_P), *Rhizobiaceae* (BS0_P), *Solirubrobacteriaceae* JAGIBJ01 (BS6_P, BP18_P), and *Rouxiella* (BS15_P) (Table S2, Fig. S3).

Regarding the citrate cycle, numerous MAGs exhibited high involvement, with 004.BP18_P standing out with a full presence. Several MAGs showed significant involvement in other pathways, such as the Entner-Doudoroff pathway and glycolysis. Similarly, MAGs exhibited varying degrees of presence in pathways like the glyoxylate cycle,

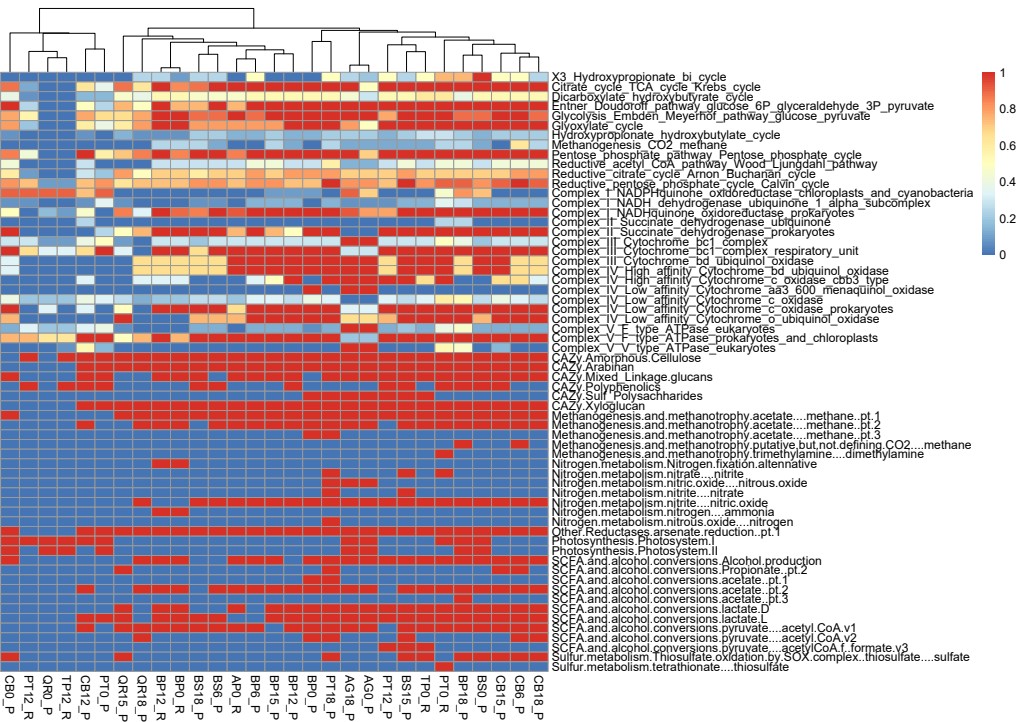

**Figure 5** Functional genes annotation of studied assemblies based on DRAM (distilled and refined annotation of metabolism).

the hydroxypropionate-hydroxybutyrate cycle, methanogenesis, the pentose phosphate pathway, the reductive acetyl-CoA pathway, and the reductive citrate cycle.

Furthermore, our functional annotation characterized the most important MAGs, revealing their involvement in key metabolic pathways associated with leaf litter degradation and energy metabolism—for example, 001.BP12_R exhibited high involvement in pathways like the reductive citrate cycle and glycolysis (Figs. 6, S3).

Moreover, our analysis suggests that microbial communities possess the genetic potential for producing volatile organic compounds (VOCs) through various metabolic pathways, including glycolysis and the glyoxylate cycle. Notably, MAGs like 002.BP0_P and 005.BP15_P (reconstructed from silver birch) showed a significant presence in these pathways, indicating their potential for VOC production.

PERMANOVA analysis reveals the substantial impact of leaf litter type and decomposition phase on functional gene diversity within microbial communities, emphasizing their dynamic influence during decomposition processes. Furthermore, functional annotation of optimized MAGs elucidates distinct metabolic profiles associated with leaf litter degradation, providing insights into the diverse metabolic capabilities of microbial taxa across different litter types and decomposition stages.

However, to advance our understanding, we should explore alternative methodologies, such as metatranscriptomics or metaproteomics, to complement metagenomics in characterizing fungal communities. Exploring multi-omics approaches can provide deeper
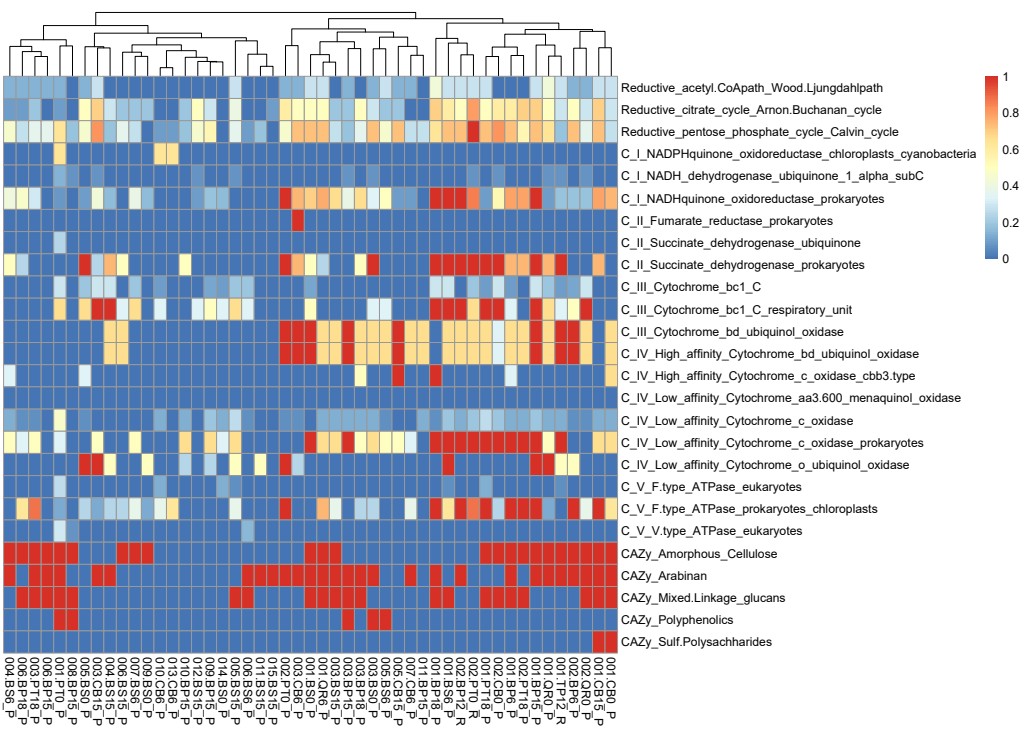

**Figure 6** Functional genes annotation of obtained MAGs based on DRAM (distilled and refined annotation of metabolism).

insights into the functional roles of fungi in leaf litter decomposition processes and ecosystem functioning.

## DISCUSSION

To our knowledge, we present the first study entirely based on metagenome analysis describing the taxonomic composition and functional characteristics of microbial communities, using samples of plant litter from temperate and boreal forests (the Bialystok region of Poland and the Perm region of Russia).

Our results revealed that the composition and structure of microbial communities changed dynamically during the 18-month experiment. There were microbial changes with the stage of decomposition and seasonality. Similarly to previous research (*Purahong et al., 2016*), we did not observe significant changes in stage-dependent shifts of bacterial communities, in contrast to the taxonomic structure of the fungal communities (Fig. 2) (*Schroeter et al., 2022*; *Purahong et al., 2016*). In our study, similarly to previously reported results (*Schroeter et al., 2022*; *Purahong et al., 2016*; *Veen et al., 2021*), there was a high abundance of *Sphingomonas* and *Rhizobacter* (birch leaf decomposing communities) but no *Nitrobacter*.

Recent work by *Schroeter et al. (2022)* is an excellent example of an experiment that proves the potential of such types of microbial communities; however, it was a short-term
experiment (2–22 days) and thus only explained the dynamics of microbial communities over a limited timeframe. Both the present research and *Schroeter et al. (2022)* showed similar results (no litter-specific association) in the dynamics of microbial communities.

The analysis of dominant taxa in studied communities revealed a high abundance of litter-decomposing taxa *Pseudomonas*, *Sphingomonas*, *Pedobacter*, and *Massilia*, as also observed by *Hu et al. (2022)* and *Tláskal et al. (2016)*. Genera *Pseudomonas* and *Massila* are characteristic of both early and later stages of decomposition that we observed in studied communities.

Abundant representation of *Aureobasidium*, with its antibacterial activities, correlated negatively with most bacterial species from the core communities.

The taxonomic composition of microbial communities across different samples highlights the dominance of specific bacterial orders, such as Burkholderiales, Sphingomonadales, and Hyphomicrobiales, in various decomposition environments (*Lu et al., 2022*). *Pseudomonas*, *Sphingomonas*, and *Stenotrophomonas* exhibit notable abundance across multiple samples, indicating their ecological significance in leaf litter decomposition (*Delgado-Baquerizo et al., 2018*; *Hu et al., 2022*; *Tláskal et al., 2016*; *Wang et al., 2018*). Additionally, diverse bacterial genera, including *Pedobacter*, *Flavobacterium*, and *Dyadobacter*, are prevalent in specific samples, reflecting the complexity of microbial interactions during decomposition (*Wang et al., 2018*).

Our results regarding the diversity of microbial communities align with previous studies, showing higher Shannon diversity index values during the later stages of birch and alder decomposition in samples from Poland, as observed by *Tláskal et al. (2016)* and *Bani et al. (2018)*.

The results of our study, corroborated by statistical test (PERMANOVA) and non-metric multidimensional scaling, emphasizing the influential role of leaf litter composition in shaping microbial community structure during decomposition (*Anderson & Walsh, 2013*; *Oksanen et al., 2013*; *Schroeter et al., 2022*; *Wang et al., 2018*). While the decomposition phase appears to have a less pronounced effect on taxonomic structure, variations in leaf litter type significantly impact microbial diversity and composition (*Schroeter et al., 2022*).

Interestingly, heat map showed a negative correlation with the metagenome distribution concerning the yeast-like fungal taxa *Aureobasidium* and the gram-negative bacteria *Pedobacter*. *Aureobasidium* spp. have been the most widely studied soil fungi and are now regarded as both effective and safe for various applications in agriculture. Indeed, these species produce a wide range of natural compounds, such as melanin, pullulan, $\beta$-glucan, aureobasidin, siderophores, glycerol-liamocin, exophilins and VOCs. Hence, *Aureobasidium* spp. is of particular interest for agriculture (*Di Francesco et al., 2015*; *Di Francesco et al., 2020*). Based on a previous study, *Pedobacter* produces fewer VOCs (*Garbeva et al., 2014*); however, particular taxa in the microbial communities can stimulate or inhibit the production of volatile chemicals by other microorganisms. Interest in VOCs has increased steadily, but publicly available data concerning the genomic bases of the compounds produced by leaf-litter-associated microorganisms is still minimal (*Lucaciu et al., 2019*).
The variations in metabolic pathways among microbial MAGs highlight the diverse functional abilities of leaf litter-degrading microbial communities (*Eichorst & Kuske, 2012*). Functional gene annotation analysis reveals that MAGs like *Brevundimonas* (001.BP15_P) also contribute to leaf litter degradation by containing genes involved in the breakdown of specific components like beta-mannan and arabinan (*Nelson, Martiny & Martiny, 2016*; *Malik et al., 2020*). Functional analysis of metagenomic data further reveals distinct metabolic profiles associated with leaf litter degradation in specific microbial MAGs. For instance, *Achromobacter* (002.PT0_R) is actively involved in pathways like the reductive citrate cycle and glycolysis, indicating its role in energy metabolism during leaf litter degradation. Additionally, our investigation suggests that microbial communities harbor the genetic potential for producing volatile organic compounds (VOCs) through various metabolic pathways like glycolysis and the glyoxylate cycle. Among the MAGs we examined, we mostly obtained bacterial MAGs. However, the fact that none of the MAGs exhibit particularly high values across both CAZy enzymes related to polysaccharide degradation and energy metabolism pathways, as seen in 001.BP15_P, classified as *Brevundimonas*, demonstrates that relatively poorly studied fungal MAGs from leaf litter seem to play a crucial role in these processes that needs to be studied further.

Despite pathway variations, functional redundancy may exist within these communities, ensuring ecosystem stability amidst environmental change (*Louca et al., 2016*). Environmental factors, such as substrate composition, moisture content, and temperature, likely influence pathway abundances, shaping the metabolic capacities of leaf litter microbial communities (*Bardgett et al., 2008*). The diverse metabolic potential observed across MAGs enhances the stability and adaptability of these communities to shifting environmental conditions (*Shade et al., 2012*; *Malik et al., 2020*).

Our analysis of metagenomic data revealed distinct metabolic profiles associated with leaf litter degradation in certain microbial taxa. For example, microbial groups exhibited significant involvement in key metabolic pathways related to energy metabolism during decomposition. Additionally, we found evidence of the genetic potential of microbial communities to produce volatile organic compounds (VOCs) through various metabolic pathways.

The result of PERMANOVA performed for functional diversity suggests that both the type and decomposition stage significantly influence functional gene diversity within the dataset, although the significance levels are slightly above conventional thresholds. These results highlight the importance of considering both factors when analyzing functional gene diversity in this context.

## Further perspectives

Further research into leaf-litter-decomposing microbial communities should focus on cultivating microorganisms, sequencing the genomes of selected taxa (bacteria and fungi) and the composition of VOCs emitted by these taxa. This study is part of a project whose purpose was to study the taxonomic and functional diversity of microbial communities decomposing leaf litter where decomposition emits VOCs. In this paper, we underline the

importance of microbial community taxonomic composition analysis to VOCs emission studies at local and global scales.

## CONCLUSIONS

Our study reveals the dynamics of microbial communities involved in the decomposition of leaf litter, highlighting the complex relationship between seasonal variations and the diversity of microorganisms. We identified specific bacterial genera associated with different stages of decomposition, suggesting specialized roles within these microbial communities. Interestingly, the functional gene composition showed less variation across seasons and litter types, suggesting a degree of stability in microbial function. Our findings highlight the diverse functional capabilities of leaf litter microbial communities and their essential roles in organic matter decomposition. Further research is needed to fully understand the contributions of microbial taxa to VOC production and their broader implications for ecosystem functioning. Including fungal MAGs and genomes in future studies is crucial for understanding the functional roles of specific taxa in litter decomposition. These insights deepen our understanding of microbial-mediated processes and highlight the importance of leaf litter composition in ecological studies.

### Funding

This research was supported by the National Science Centre (Poland), grant number 2019/35/B/ST10/02252. The funders had no role in study design, data collection and analysis, decision to publish, or preparation of the manuscript.

### Grant Disclosures

The following grant information was disclosed by the authors:
National Science Centre (Poland): 2019/35/B/ST10/02252.

### Competing Interests

The authors declare there are no competing interests.

### Author Contributions

- Nataliia Khomutovska conceived and designed the experiments, performed the experiments, analyzed the data, prepared figures and/or tables, authored or reviewed drafts of the article, and approved the final draft.
- Iwona Jasser conceived and designed the experiments, authored or reviewed drafts of the article, and approved the final draft.
- Polina Sarapultseva performed the experiments, authored or reviewed drafts of the article, field experiments with litter and preparation for microbiological research, and approved the final draft.
- Viktoria Spirina performed the experiments, authored or reviewed drafts of the article, field experiments with litter and preparation for microbiological research, and approved the final draft.

- Andrei Zaitsev conceived and designed the experiments, authored or reviewed drafts of the article, supervising the Russian part of the project, and approved the final draft.
- Jolanta Masłowiecka performed the experiments, authored or reviewed drafts of the article, field experiments with litter and preparation for microbiological research, and approved the final draft.
- Valery A. Isidorov conceived and designed the experiments, authored or reviewed drafts of the article, and approved the final draft.

## Data Availability

The Illumina NovaSeq raw reads are available at BioProject: PRJNA1001592.

## Supplemental Information

Supplemental information for this article can be found online at http://dx.doi.org/10.7717/peerj.17769#supplemental-information.

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
