# Peer review of "Seasonal dynamics in leaf litter decomposing microbial communities in temperate forests: a whole-genome- sequencing-based study"

_PeerJ, doi:10.7717/peerj.17769_

## Round 0.1 · original submission · Major Revisions

The manuscript in this current state is not suitable for publishing. Please make the suggested changes, and the editorial process will continue.

**Language Note:** The review process has identified that the English language must be improved. PeerJ can provide language editing services - please contact us at [email protected] for pricing (be sure to provide your manuscript number and title). Alternatively, you should make your own arrangements to improve the language quality and provide details in your response letter. – PeerJ Staff

Reviewer 1 ·

Excellent Review

This review has been rated excellent by staff (in the top 15% of reviews)
EDITOR COMMENT
The reviewer did an excellent job. The report was thorough and respectful, and I am convinced that it will help the authors a lot to improve their manuscript. I want to thank them very much.

Basic reporting

1. The manuscript is written in a clear, concise language. Though it seems that introduction, results and discussion are written by separate people - while introduction states about the significance of the analysis of a decomposing community functioning through metagenome sequencing, the discussion tells about the significance of the taxonomic analysis.

2. The introduction starts well, but it is a little bit all over the place, maybe because the research didn't state any clear purpose. The authors start with the importance of understanding the functioning of natural ecosystem, then authors switch to volatile organic compounds (VOC), which is a very narrow theme. Later they refer to them as "volatile secondary metabolites", which puzzles the reader. Please choose and use one term.
Then authors briefly mention the problem of DNA extraction. Indeed, quality of DNA isolation is a good research topic, but was it yours? It does not seem like this after the reading of the whole manuscript. In my opinion, it distracts the reader.
Then authors write about well-represented research on litter degrading fungi in order to justify the focus of the bacterial component. But in the results authors speculate quite a lot about fungi taxonomy. Why, if you decided to focus on bacteria?
Next authors dwell on the importance of metagenomic approach, which is unarguable. But later they mentioned that "the functional potential of a community (functional gene analysis) has not been explored", which is totally not true. There is a lot of research on functional annotation of degrading communities, including leaf litter (eg Nelson 2015, 10.1128/AEM.02222-15; Malik 2020, 10.1038/s41396-020-0683-6; Taketani 2018, 10.7717/peerj.5710; Guerreiro 2023 10.1007/s11557-022-01859-0).
Another arguable statement by the authors is that degrading community is an example of plant-microbial interactions and they even compare it to the mycorrhiza. Well, my gut feeling is both partners in the evolving interaction should be alive. But in the matter of decomposition, plant material is already dead and does not evolve.
Next authors introduce the type of litter as an important factor of microbial community composition, which is a good hypothesis to check. Then they explain it by the secondary metabolites, synthesized during decomposition. But these metabolites are secreted by decomposers, not the substrate. So technically it is the auto regulation of the microbial community. Again, the idea is understandable, but the whole paragraph is written in a confusing way.
The hypothesis and the objectives of the study sound good, the methods used are really good, but the data analysis and results presentation are very poor.

3. The article structure is ok.
Figures are relevant to the text, but of inadequate quality. Samples from different litter types, seasons or regions are hard to distinguish. Figures 2-4 are excessive and focus on the taxonomic composition rather than aforementioned functional annotation.
Not all data, mentioned in the results, is presented. The nomenclature of bacterial phyla is old.

4. The submission is self-constrained, but results are not relevant to the hypothesis.

Experimental design

The manuscript describes an original research, which, if presented correctly, would be a great addition to the journal. However, in the current state its presentation is not suitable for publication.
1. Experiment design is vague and not explained precisely. The core of the research is the dynamics of leaf litter decomposition, which is very promising. But it is not clear, why authors use two sampling sites - in Poland and Russia. I don't see justification neither in introduction, nor in the methods section. What is the reason of including these two completely different areas? The results do not explore this question.
The description of the experiment design is not sufficient. When was the leaf material collected, how it was treated before putting in the bags? The time of experiment layout is also not clear. Figure S1 shows only experiment set up from Poland. Where is the figure for Russia? Please modify the experiment description accordingly.
The part about mock community is the most confusing. According to the results section, it plays a great role in the research. Please add the description of experiment design concerning mock community. Or move it entirely to the different paper, because I don't see its significance for the current research.
The analysis of the data described not sufficiently and do not cover all methods used in the results section (eg Shannon, PCoA, Antismash, SEED)

Validity of the findings

The sequencing of 30 metagenomes is a very promising amount of data to test the hypothesis stated in the introduction. However, I find the analysis quite poor and lacking interest to the reader. Most of the result section covers taxonomic composition of decomposing community, though originally it aimed at the functional annotation. The functional annotation itself is very brief and not illustrated sufficiently. The Subsystems classification is very general and does not provide sufficient resolution to look into the differences between different seasons or litter types. I would suggest enrichment analysis to look into these differences on the KO terms level. Also, when it comes to the decomposition community, it is very helpful to dig into CAZy database, which provides the classification of GH genes, responsible for the degradation of lignocellulosic substrates.
The part with secondary metabolites is the most confusing. Why were they studied in the MC sample? The planning and structure of the research is very confusing at this moment.
The description of PCoA plot without any statistical confirmation is not enough to state any shifts of bacterial community.
The discussion should also more closely resemble the findings of the paper. In its present form it tells a lot about taxonomy composition of the decomposer community, while initially the paper wanted to focus on the function of the community.
In conclusion, the research is potentially very valuable for the field, the experiment and the data are very promising and can be a material for a great article, but the manuscript in its present form can't be published. You need to reevaluate your hypothesis, the design of the experiment (specifically two locations and mock community) and the results, which you include and align it with the discussion and the conclusion. The analysis of metagenomes is too brief and does not reveal its potential.

Additional comments

Here are more specific notes:

Line 29 please update bacteria nomenclature here and in all the manuscript to the modern standards (Proteobacteria=Pseudomonadota, Actinobacteria=Actinobacteriota, etc) (Oren&Garrity, 2021, https://doi.org/10.1099/ijsem.0.005056)
Line 42 «Despite …» Please check the meaning of the sentence
Line 46 May be add information about VOC in the abstract
Lines 84-90 Why do you propose that decomposing leaf litter is an example of coevolving plant-microbial associations? To my understanding, these include interactions between living organisms, like PGPR or rhizobia, not living decomposing the dead, which are merely considered copiotrophs
Line 99 - who produces the chemicals?
Line 101-102 Unexpected introduction of seasonal changes, though the passage was mostly about secondary metabolites of bacteria
Line 104 - Functional annotation of leaf-decomposing communities have been explored before Nelson 2015, 10.1128/AEM.02222-15; Malik 2020, 10.1038/s41396-020-0683-6; Taketani 2018, 10.7717/peerj.5710; Guerreiro 2023 10.1007/s11557-022-01859-0
Line 145 How did you collect and prepare leaf litter used in the bags? Was it sorted, sterilized, grounded? How many bags for each plant variant?
Line 149 At what season was the experiment conducted? It is not clear how long were the samples incubated.
Line 153 - were samples dried or frozen? Which ones which?
Line 159 - do you refer to «fresh» samples as the least incubated or frozen?
Line 171 - What actual analysis of metagenome data was conducted?
Line 184-188 The usage and preparation of mock community should have been mentioned and explained earlier.
Line 190 - Figure 2 does not carry information about the total number of sequences
Line 233 - Why on earth here appear the «16S results»?
Line 234 - Methods didn’t mention anything about alpha-diversity calculation. Please add this information
Line 238 - The PCoA plot based on the NCBI database - add information to the methods
Line 245 - how PCoA indicates gradual changing of the community?
Line 248 The PCoA based on the Bray-Curtis distance matrix
Line 256 - the neighbour-joining comparison
Line 271 - Where exactly is your functional annotation of leaf-decomposing communities?
Line 278-283 - where is the data?
Line 285 - where was it stated before that you were going to analyze secondary metabolites of MC?
Line 375-377 why do you put here that you wanted to underline the importance of the microbial communities' taxonomic composition analysis, while earlier you were pointing out the importance of functional full-metagenomic analysis?

Reviewer 2 ·

Basic reporting

Please refer to "additional comment" report

Experimental design

Please refer to "additional comment" report

Validity of the findings

Please refer to "additional comment" report

Additional comments

The manuscript presents a comprehensive investigation into the taxonomic composition, dynamics, and functional roles of microbial communities involved in leaf litter decomposition (as part of the global carbon cycle) in European temperate forests across different ecological conditions. The study employs metagenome sequencing and metagenome-assembly-based approaches for evaluating the microbial communities associated with the leaf litter of eight forest-forming species of woody plants. The results showed that despite the microbial community varies depending on the stage of litter decomposition, Proteobacteria (Alpha- and Gammaproteobacteria), Actinobacteria, Bacteroides, and Ascomycota (Leotiomycetes) are consistently detected as the core microbial community during this process. The authors also analysed the gene clusters related to volatile secondary metabolites biosynthesis associated with litter degradation, adding a functional dimension to the study.
Although the paper could address important ecological questions and promise a valuable contribution to the field, some improvements are necessary to clearly state the context in which the study is conducted, particularly with the identification of clear hypothesis(es)/aim(s) already in the abstract and integration of findings and discussion.

Strengths
- Whole-genome sequencing provides a robust foundation for understanding the taxonomic and functional diversity of microbial communities involved in leaf litter decomposition.
- The inclusion of multiple forest-forming tree species and the assessment of microbial communities in two European regions in multiple seasons contribute to the generalizability of the findings. However, the experimental design and the tested factors must be clearly stated. Which are the comparisons done here? What do the authors want to prove? Statistical analysis is not provided to support the findings.
- Even if the manuscript underscores the importance of unravelling the intricacies of microbial communities involved in leaf litter decomposition for advancing our understanding of ecosystem processes and the carbon cycle, especially if we consider ongoing climate change, it lacks a straightforward experimental design and statistical approach/replication to support/answer the state questions. The authors should address it to corroborate the work and findings presented in this work.

Comments and Integration of findings:
- Line 25. I would use the term metagenome-assembled genomes (MAGs) to indicate the different microorganisms obtained/reconstructed from metagenomes (please clarify it also in the methods). I would remove the word “isolation” because metagenomes were obtained from sequencing starting from sample DNA. Please rephrase. As well, why “cultivated” term was used? From what is reported in the methods, all work is based on the sequencing of DNA extracted from litter. Please revise.
- Enhance the integration of taxonomic and functional findings throughout the manuscript. Discuss how changes in taxonomic composition correspond to functional shifts, especially during different stages of litter decomposition.
- Discussion of Implications. Extend the discussion section to include a more thorough exploration of the implications of the findings for broader ecological processes, considering the potential impacts on nutrient cycling, ecosystem resilience, and carbon sequestration.
- The introduction does not present a clear background and “ecological concepts/perspective” supporting the hypotheses. It lists a series of statements, for which of them what is still missing is underlined. I suggest modifying the flow and providing more justification for your study based on an “ecological” perspective. For instance, why should season change be more significant than litter composition? Is it related to similar composition of leaves or drastic changes across seasons regarding temperature, water availability, etc.? In line 108, only a list of references is reported without clearly stating why/how the authors arrived at their hypothesis. What microenvironment is? Do you mean the “litter niche” or other factors/conditions?
- Line 112. I find this statement contradictory with the previous: if environmental conditions are stronger shapers than litter composition, why do you expect a dominant litter “core microbiome” across different forests? Is this related to the concept of “litter” as a micro-environment (niche) that strongly selects the associated microbiome? Please clarify it.
- How do you consider the effect of climate/environment? Is it a selective force?
- I do not understand the combination of these two hypotheses in this order. Is the litter or the season the main selective factor proposed here? I think that litter can be considered the first factor (see second hypothesis) since it gives the C-source and component necessary for microbial metabolisms and, thus, the development of the community. However, the fact that the litter is also exposed to the different climate/environmental conditions of the forests (e.g., seasonality) introduces an additional selective force that can further affect/tune the microbial diversity/functionality.
- Line 123. Did you succeed in assembling fungi from metagenome? Which portion was obtained? If the work is focused on bacteria only, I would remove the part related to fungal roles or clarify that only certain functions were explored due to limitations in fungal sequencing.
- Please add representative pictures of the litter/forest sampled for this study, including those during different seasons; in Sup Fig. 1, only one sampling site is reported.
- Line 144. Please clarify the aim of the “incubation” step applied to litter (Sup Fig 1). Please provide a caption/ explanation for this Sup. Fig. 1, sorry but I didn’t find it.
- The manuscript presents too many figures; they can be assembled into fewer figures with multi-panels.
- Figure 1 is not visible (the grey shade covers the colour of the leaves). If leaves were not detected in a specific forest, please remove them or apply transparent/shaded colour to differentiate them from those present. Change the white of the winter with another colour. Why are there five lines? Are these decomposition levels? If yes, how does this reflect seasonality? Please clarify the figure and add all the details to the caption.
- Figures 2, 3 and 4. Many labels are not readable; use a bigger font size. Combine these two in a unique figure with A, B and C panels. Are these results from MAGs? Or from taxonomic identification of specific genes? Please clarify.
- Figure 3. It states that the entire community is reported; please clearly indicate which are bacteria and fungi. Labels cannot be written, so it is difficult to understand them in their current form.
- Figure 5. Add labelling of the axes (diversity explained) and colour legend to understand sample distribution in the ordination. Samples are intermixed without an apparent clustering; how do you explain it? Does the season/litter not significantly affect the community? How core microbiome cluster in the ordination? The core microbes have different distributions across samples (see Figure 4), and it would be interesting to see how this portion of the community ordinates in the space. Which genes were used to run such an analysis at the species level? How is reads/abundance calculated across samples/species?
- Figure 6. What nodes are? And their colour? Are MAGs? Why are some nodes not connected to any others (no edge/degree) but are included in the network? Add more details in the caption; otherwise, it is difficult to understand the figure.
- Figure 8. From which MAGs? How many microbes possess this cluster? Please add the necessary details to understand the figure and add the presence of these clusters across samples. Moreover, I suggest merging this figure with Figure 7 as two separate panels.
- Line 132. Why are environmental data reported only up to 2007? Temperature/rain datasets are also available for more recent years.
- Provide additional sequencing and analytical methods details, ensuring transparency for readers seeking to replicate or build upon the study. Among others, report how the library was prepared and explain the mock community used as a control in the methods (not only in the results). More details on reads obtained, MAGs reconstruction, and metagenome analysis should be reported. How many plant reads were discarded? Are the remaining reads enough to “describe” the community?
- What means the host DNA was degraded? How was this checked?
- Which kind of statistical analysis was applied? And with which experimental design? Factors tested and the type of stat tested should be clearly stated. For instance, how were differences in taxonomy evaluated/tested? Where are the values of significance (p-values) reported?
- Same for the ordination. It does not have statistical values; additional analyses (e.g., PERMANOVA, adonis, manyglm) should be provided to support similarity or not among tested groups. Clearly state the number of replicates per group.
- How was network analysis performed? No info is reported in the methods. What is the aim of running it? Which samples/MAGs were included and why?
- Please carefully revise methods and results to be sure all info/details is/are reported. In this form, it is difficult to follow the results and understand why specific data are presented.

---

## Round 0.2 · Major Revisions

I agree with the first reviewer that we must improve the manuscript before accepting it.

Reviewer 1 ·

Basic reporting

The manuscript had undergone a major revision and was significantly improved from the previous version.
However, I still have some concerns about the text.

1. The introduction was updated according to the recommendations and has more clear structure, but I think that there is a lot of repetition in lines 122-149. You repeat your hypothesis twice, which is not necessary. Please rewrite to make it more concise.

3. While I appreciate the updated Figure 1, I think that materials and methods section should provide the information about the type and quantity of collected samples, including plant species and the period of decomposition.

4. The results section has been reworked, but somehow it became worse. The monotonous listing of taxa found in 29 samples is totally unreadable (lines 233-311). First of all, since you have many plant species, it is very difficult to follow all the sample names. Plus, some of the read numbers in brackets seem to be inappropriate (e.g. line 309), which suggests that list of samples can also contain mistakes. Please remove this part and leave small concise description, highlighting the differences between region/plant species/decomposition state.

5. Furthermore, figures 2-4 are still redundant. They are big, take up a lot of space and don't present a clear message. Leave one figure and the others move to the supplement. I also recommend trying to highlight different features of samples - region/plant species/decomposition state. Additionally, the figure description should have the explanation of sample IDs.

6. Clade usually refers to a part of a phylogeny. But on lines 316-321 you refer to clades on Figure 5. Figure 5 does not contain a phylogeny. Did you mean "cluster"? If so, please highlight clusters on this figure and add this information to the figure description.

7. You write on the line 322 "The conclusions drawn from our analysis align closely with the results of the Adonis (PERMANOVA) test." But you didn't highlight any conclusions from your analysis before this sentence. Ideally, each paragraph should have the description of results obtained from a certain analysis and be finished with a sentence concluding the findings by this analysis.

8. Functional gene analysis description has been improved, but also has flaws similar to taxonomic analysis. Lines 334-347 would benefit from some structurization of the findings. How did you determine "the highest diversity of metabolic pathways"? How did you determine that "BP18_P is similar to PT0_R,"? They are not the closest to each other on the dendrogram on figure 6.

9. Description analysis of MAGs on line 353 should start with basic reporting from lines 370-372 - number of identified MAGs, completeness, etc.

10. In the discussion how do you distinguish between seasonal and decomposition stage driven changes in the microbiome?
In line 412 you mention "Principal coordinate analysis elucidates distinct clustering patterns", but according to your materials and results section, you didn't use this analysis.
Lines 419-425, 463-465, 478-482 do not discuss the findings, they may be included in the conclusion, but I don't see the significance of these phrases in the discussion.
Lines 457-463 contain obvious information. It would be more novel if you elaborate which microorganisms (taxonomic attribution of these MAGs) contained these genes.

Experimental design

In the line 185 you mention that leaf litter samples were dried, but in the line 188 you say that it was frozen. Which statement is correct?
Lines 210 and 215 - no need to repeat MAGs abbreviation decryption
I find that "We employed stringent quality control measures and bioinformatics analysis techniques to maximize the accuracy and reliability of our data" sentence in lines 218-220 is unnecessary.

Validity of the findings

Most of the findings are quite weak and repeat the previous research. Authors too often repeat themselves that their "findings underscore the importance of microbial communities in leaf litter degradation". Please minimize the stating of the obvious truths, like "For instance, 001.BP12_R is actively involved in pathways like the reductive citrate cycle and glycolysis, indicating its role in energy metabolism during leaf litter degradation.". It's good that you found a MAG complete enough to have an energy pathway, but most of the organisms contain energy pathways in their genomes.

Additional comments

Line 233 - "According to the NCBI database, 2,820,491 assigned sequences passed quality filters (Figure 2)." - Figure 2 does not contain this data.

Reviewer 2 ·

Basic reporting

Despite the manuscript being clearer, several parts are still difficult to follow, while figures can be combined or moved in SM (there are so many heat maps). Moreover, the authors stated that "All modified chapters are now clearly marked in yellow"; however, the file did not show any highlights but only track changes, making the revision process more complicated, especially if the entire parts of the text are cancelled and rewritten/moved.
I suggest that the authors could provide files in which revision can be better evaluated, as well as work on the "visualisation" of results.

Experimental design

The authors clarified it.

Validity of the findings

Potentially interesting, but still not clearly reported/discussed

---

## Round 0.3 · accepted · Accept

Please make a last effort to polish the many many details that are still missing for the final publication, both reviewers offer good advice.

Reviewer 1 ·

Basic reporting

By the third round of the review the manuscript didn't undergo pivoting changes.
It still have ways to be improved, but at this point I'm out of ideas.
I understand that taxonomic information is easier to process than functional metagenomics, but I woul reccomend authors to pay more attention to the latter in the future. The search of functional differences is a challenging process, which requires a lot of data and replicates, which is not always readily available.

Here are some minor comments:
Please add specificity of your findings concerning metabolic profiles associated with leaf litter degradation in the abstract.
Line 286 – please provide the results of PERMANOVA.
Line 293 – there is no need to clarify that “a p-value of 0.566 is exceeding the significance level of 0.05”.
Line 276- please provide the information about the measurement of Shannon diversity index values in the methods section.
Please remove figure 2 from the main text of the manuscript. The shift from the heatmap to the MS excel histogram didn't make it prettier.

Experimental design

-

Validity of the findings

-

Additional comments

-

Reviewer 2 ·

Basic reporting

The authors addressed the comments of reviewers, making the text more clear. They added the necessary details to the methods and revised both the intro and discussion. However, captions and figures still miss info/explanation fundamental to understanding the data reported/shown (e.g., among others, in heat maps, what does the scale indicate? Are these lo transformed values? or relative abundance?). Please provide all details necessary to understand the figure and all code/abbreviation reported (without relay on the info reported in the text of the manuscript), as well as the type of data plotted and the stat supporting them (when possible).

Experimental design

See above

Validity of the findings

See above

Additional comments

See above